

# Effects of exogenous zinc on the physiological characteristics and enzyme activities of *Passiflora edulis* Sims f. *edulis* seedlings

Jianli Zhang[1], Tao Yang[1], Chen Zhang[1], Ting Zhang[1], Lihua Pu[1] and Weiquan Zhao[2]

[1] College of Eco-Environmental Engineering, Guizhou Minzu University, Guiyang, Asia, China
[2] Institute of Mountain Resources of Guizhou Province, Guizhou Academy of Sciences, Guiyang, Asia, China

Corresponding author
Jianli Zhang, zhangjl-z@163.com

## ABSTRACT

Passionflower (*Passiflora edulis* Sims) is widely distributed in tropical and subtropical areas for edible, medicinal and skin care product processing, and the market demand is large. Zinc (Zn) is a necessary trace element for plant growth and development. In many countries, the content of Zn in soil is low and/or bioavailability is low. The exogenous application of Zn has become a common agronomic measure in agriculture. However, the effect of Zn on the physiological characteristics and enzyme activity of passionflower seedlings is not clear. In this study, pot experiments were conducted to analyse the effects of different concentrations of Zn (0, 200, 400, 800 mg kg$^{-1}$) on the plant growth, photosynthetic pigments, osmotic regulators, membrane system and antioxidant enzyme system of purple passionflower (*Passiflora edulis* Sims f. *edulis*) seedlings, and Pearson correlation and principal component analyses were performed. The results showed that (1) the 200 mg kg$^{-1}$ Zn treatment increased the contents of chlorophyll a (37.65%), chlorophyll b (41.22%), chlorophyll a+b (38.59%) and carotenoids (29.74%). The value of chlorophyll a/b changed little and had no effect on leaf growth. (2) The contents of proline (Pro) and malondialdehyde (MDA) in *P. edulis* Sims f. *edulis* seedlings treated with 400 mg kg$^{-1}$ Zn increased significantly by 116.84% and 42.69%, respectively. The activities of catalase (CAT) and peroxidase (POD) increased by 16.82% and 18.70%, respectively. Superoxide dismutase (SOD), leaf area (LA), leaf perimeter (LP) and leaf width (LW) decreased significantly by 47.20%, 19.75%, 8.32% and 11.97%, respectively. (3) 800 mg kg$^{-1}$ Zn significantly increased the contents of Pro (202.56%) and MDA (26.7%) and the activities of CAT (16.00%) and POD (67.00%), while the soluble sugar (SS), SOD, LA, LP and LW decreased significantly by 36.67%, 32.86%, 23.36%, 8.32% and 11.18%, respectively. (4) There was a significant positive correlation between Pro and photosynthetic pigments and between SOD and leaf growth and a significant negative correlation between POD and SS and between SOD and MDA. (5) A low concentration (200 mg kg$^{-1}$) of Zn promoted the growth of *P. edulis* Sims f. *edulis* seedlings and allowed stress caused by high Zn concentrations to be tolerated. The results of this study can provide a reference for the application of Zn fertilizer to *P. edulis* Sims f. *edulis*.

# INTRODUCTION

Zn is one of the essential trace elements for plant growth and development and plays an important role in plant growth (*Gharaibeh et al., 2016*; *Shuai et al., 2022*). Zinc can be used as an auxiliary factor in six kinds of functional enzymes in plants and regulates the activity of enzymes (*Stanton et al., 2022*). Zinc is considered an important element in a series of physiological processes of plants, such as photosynthesis, protein synthesis, antioxidant function, pollination and growth regulation (*Pavia et al., 2019*). Zn is beneficial for maintaining the integrity and stability of the cell membrane (*Sarwar et al., 2015*). The low content of Zn and/or low bioavailability in the soil of many countries in the world leads to a lack of Zn in crop (*Prasad et al., 2012*; *Zhao et al., 2018*). Zinc deficiency causes a decrease in the chlorophyll content and photosynthesis of crops, destroys the metabolic balance of reactive oxygen species (ROS) in crops, and causes oxidative damage to the cell membrane, thus reducing crop yield (*Hajiboland & Amirazad, 2010*). Appropriate amounts of Zn can activate the expression of ROS-related detoxification genes and reduce the toxicity of ROS (*Subramanian et al., 2011*). Therefore, it is of great significance for crop growth to supplement Zn in Zn-deficient soil, and the exogenous application of Zn has become a common effective measure to increase the soil Zn content (*Zhao et al., 2018*). $ZnSO_4$, as a common Zn fertilizer, has been used in fruit cultivation to improve fruit yield and quality (*Safa et al., 2020*).

It has been reported that the foliar spraying of Zn can promote the growth, chlorophyll content and photosynthesis of coffee trees (*Rossi et al., 2019*). Zn fertilizer can also effectively improve the activities of disease resistance-related enzymes in long-term continuously cropped soil potato roots, reduce the content of malondialdehyde (MDA), and reduce the occurrence of soil-borne diseases (*Xie et al., 2022*). In addition, the spraying of zinc oxide nanofertilizers (ZnO NPs) on corn or soil can also achieve the above effects (*Azam et al., 2022*). The exogenous supplementation of Zn can also effectively alleviate stress in plants, such as drought, saline-alkali and heavy metal stress (*Ali et al., 2022*; *Pavia et al., 2019*; *Ramezanian et al., 2010*). *Wei et al. (2022)* showed that a low concentration of Zn increased the chlorophyll content and antioxidant enzyme activity in wheat, while a high concentration of Zn promoted MDA production and cell membrane permeability. This shows that an appropriate amount of Zn is beneficial to the synthesis and stability of plant chlorophyll (*Hajiboland & Amirazad, 2010*), but excessive Zn will destroy the structure of the plant cell membrane, cause oxidative damage and interfere with the normal physiological activities of plant cells (*Li, Zeng & Su, 2022*). Excessive Zn will also lead to the accumulation of the toxic heavy metals cadmium (Cd) and lead (Pb) in plants, which is disadvantageous to plant growth and development and human health (*Stanton et al., 2022*). In summary, a reasonable application of Zn is beneficial for crop growth, yield and quality.

*Passiflora edulis* Sims is an herbaceous vine of the genus *Passiflora* L. of the *Passifloracea* family, also known as passion fruit. It has medicinal and nutritional value and is one of the favourite fruits (*Fonseca et al., 2022*; *Lozano-Montaña et al., 2021*; *Yan et al., 2021*). *Passiflora edulis* Sims is native to Brazil and is distributed in tropical and subtropical regions (*Liu, Tang & Zhou, 2021*). *P. edulis* Sims is grown in many places in southern China and the market demand has increased rapidly in recent years (*Wang et al., 2023*). As one of the main cultivated varieties, *P. edulis* Sims f. *edulis* has an important impact on the economic development of southern China. As a necessary mineral nutrient element for plants, Zn has an important effect on the growth and physiological functions of crops (*Gupta, Srivastava & Singh, 2023*). At present, research on the application of Zn fertilizer is mainly aimed at wheat (*Li, Zeng & Su, 2022*), rice (*Yang et al., 2021*) and other crops, and research on *P. edulis* Sims is mainly focused on the prevention and control of viral diseases (*Wang et al., 2023*), cultivation and breeding (*Phong et al., 2022*). There are few reports on the application of Zn fertilizer to *P. edulis* Sims f. *edulis* seedlings, which is not conducive to the popularization and utilization of Zn fertilizer in the efficient production of *P. edulis* Sims. Therefore, it is very important to study the effects of different concentrations of Zn on the physiological characteristics of *P. edulis* Sims f. *edulis* seedlings.

In this study, a suitable variety of *P. edulis* Sims f. *edulis* in Guizhou was used as the research object, and different concentrations of Zn were added to the potting soil. The main goals were (1) to explore the effects of different concentrations of Zn on the growth of *P. edulis* Sims f. *edulis* seedlings, and to explore the appropriate threshold of Zn application, (2) to analyse the effects of different concentrations of Zn on the physiological characteristics and enzyme activity of *P. edulis* Sims f. *edulis* seedlings and (3) to study the relationship between the growth and physiological changes of *P. edulis* Sims f. *edulis* seedlings supplemented with Zn.

## MATERIAL AND METHODS

### Experimental design

A 2-month pot experiment was carried out at the experimental site of Guizhou Minzu University (University City Campus) in Huaxi District, Guiyang City, Guizhou Province, China ($106°37'25.08''$ E, $26°22'32.43''$ N). The experimental area has a typical highland monsoon humid climate, with sufficient annual rainfall, an annual average temperature of 15.7 °C, an extreme maximum temperature of 34 °C, and an extreme minimum temperature of $-3.1$ °C. *P. edulis* Sims f. *edulis* seedlings were used as experimental materials. Soil from which stones and grass roots were removed was added to a plastic basin with a height of 31.5 cm and inner diameter of 48 cm, and the soil was evenly turned (tested soil: pH 6.90, organic matter 190.57 g kg$^{-1}$, cation exchange capacity 161 mg kg$^{-1}$). Well-growing and uniform passionflower plants were transplanted into a large pot and adapted to culture for 7 days. Exogenous Zn was uniformly added to the soil in the form of a $ZnSO_4$ $7H_2O$ aqueous solution. Studies have shown that the effectiveness of $ZnSO_4$ $7H_2O$ in soil can persist for 200 d (*Zhao et al., 2011*), so $ZnSO_4$ fertilizer was effective in this experimental cycle. According to the soil Zn background value of 99.5 mg kg$^{-1}$ in Guizhou

Province (*Ma et al., 2019*), the Zn concentration concentrations were set to 200, 400 and 800 mg kg$^{-1}$ (calculated by soil dry weight). Each treatment was repeated three times, there was one plant per pot, and a blank control (CK) was set. During the experiment, water was added to keep the soil moisture consistent, and 1‰ urea was applied every 7 days to maintain plant growth. A transparent plastic cover was used to seal the top, and holes allowed for the plants to pass through while reducing the leaching of rain water. There was a diversion tube and a kettle at the bottom of the basin, which was checked regularly, and water was poured back into the pot to reduce the loss of Zn.

### Index determination

The fifth leaf from the top of the passionflower plants with good growth was removed, and the leaf structure index was measured immediately by a Yaxin-1241 leaf area meter (Beijing Yaxin Liyi Technology Co., Ltd., China). The leaf area (LA), leaf perimeter (LP), leaf length (LL) and leaf width (LW) were recorded, and then the photosynthetic pigment content was determined. A total of 0.2 g of chopped fresh leaves was weighed, added to a small amount of quartz sand and an appropriate amount of 95% ethanol, ground into a homogenate, filtered into a 25 mL brown volumetric bottle and fixed in 95% ethanol. The chloroplast pigment extract with a constant volume was transferred to a quartz colorimetric plate, and 95% ethanol was used as a blank. The absorbances at wavelengths of 665 nm, 649 nm and 470 nm were measured, and the chlorophyll a (Chl a), chlorophyll b (Chl b), chlorophyll a+b (Chl a+b), carotenoid (Car) contents and chlorophyll a/chlorophyll b (Chl a/b) values were calculated. The following equations were calculated (*Cai, 2013*):

$$C_a(mg\ L^{-1}) = (13.95A_{665} - 6.88A_{649}) \qquad (1)$$

$$C_b(mg\ L^{-1}) = (24.96A_{649} - 7.32A_{665}) \qquad (2)$$

$$C_x(mg\ L^{-1}) = (1000A_{470} - 2.05C_a - 114.8C_b)/245 \qquad (3)$$

$$\text{Photosynthetic pigment content}(mg\ g^{-1}) = (C \times V \times N)/(W \times 1000). \qquad (4)$$

In the above equations, $C_a$, $C_b$ and $C_x$ denote the concentrations of Chl a, Chl b and Car, respectively, in the extracts, C is the corresponding pigment concentration (mg L$^{-1}$), V is the volume of the extract (ml), N is the dilution multiple, and W is the fresh mass of the sample (g).

The stretched leaves of passionflower plants with the same tip and good growth in different treatments were used to determine the biochemical indexes immediately after extraction. The contents of malondialdehyde (MDA), proline (Pro), soluble sugars (SS) (*Li, 2000*), superoxide dismutase (SOD), catalase (CAT) and peroxidase (POD) were determined by the thiobarbituric acid method, ninhydrin colorimetric method, anthrone sulfuric acid method and enzyme labelling method. The kits were purchased from Suzhou Mengxi Biomedical Science and Technology Co., Ltd. The SOD kit refers to the method of *Ukeda et al. (1999)*, the CAT kit refers to the method of *Peng et al. (2009)*, and the POD kit refers to the method of *Wang et al. (2014)*. The steps for CAT, POD and SOD activity determination were the same: 0.1 g fresh sample was weighed in a mortar, one mL extract was added followed by grinding into a homogenate in an ice bath, centrifugation (12000 r

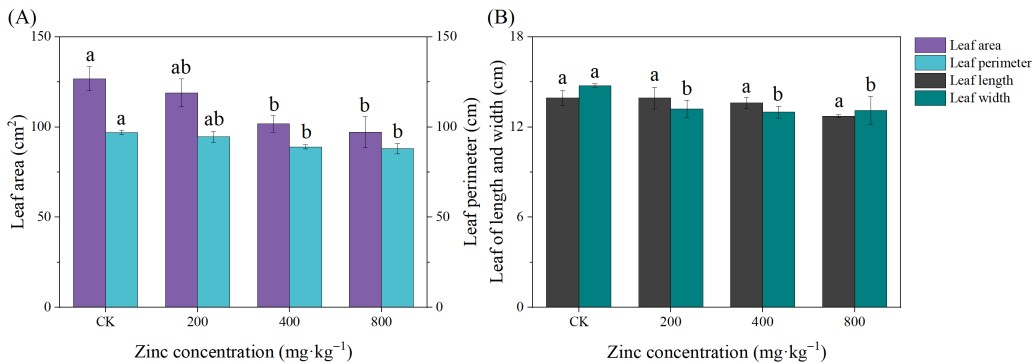

**Figure 1** **Effect of exogenous Zn on the leaf growth of *P. edulis* Sims f. *edulis* seedlings.** Different lowercase letters indicate significant differences between treatments ($P < 0.05$). The same below.

min$^{-1}$ at 4 °C for 10 min) was conducted, the supernatant was placed on ice to be tested, the reagents were added according to the method in the kit and then placed in a 96-well plate, the absorbance was determined by a Thermo Scientific Multiskan FC enzyme labelling instrument, and the enzyme activity was calculated.

## Statistical analysis

Excel 2016 was used to collate the data, and SPSS 26.0 was used for one-way ANOVA and Pearson's correlation. The significance of multiple comparative differences was analysed by the Duncan method. The data that did not satisfy the normality and homogeneity of variance were tested by the Mann–Whitney method. OriginPro 2021 was used to produce the graphics. The data in the charts are the mean ± standard error.

## RESULTS

### Plant growth

Leaves are the main organs of plant photosynthesis and can also reflect the growth status of plants. As shown in Fig. 1, the leaf growth of passionflower was inhibited with increasing Zn concentration. Except for leaf width, the addition of 200 mg kg$^{-1}$ Zn had no significant effect on the structural indexes of the *P. edulis* Sims f. *edulis* seedling leaves ($P > 0.05$). The addition of 400 mg kg$^{-1}$ and 800 mg kg$^{-1}$ Zn significantly inhibited the growth of the *P. edulis* Sims f. *edulis* seedling leaves; the LA decreased by 19.75% and 23.36%, the LP decreased by 9.26% and 8.32%, and the LW decreased by 11.97% and 11.18% ($P > 0.05$), respectively.

Chl and Car are important photosynthetic pigments in plant leaves and indispensable substances for plant photosynthesis (*Tie et al., 2020*). As shown in Table 1, exogenous Zn effectively promoted the synthesis of Chl and Car in *P. edulis* Sims f. *edulis* seedlings, which is beneficial to photosynthesis. Compared with CK, the addition of 200 mg kg$^{-1}$ Zn significantly increased the contents of Chl a (37.65%), Chl b (41.22%), Chl a+b (38.59%) and Car (29.74%) ($P < 0.05$), and the value of Chl a/b decreased significantly by 2.46% ($P < 0.05$). The contents of Chl and Car increased with the addition of 400 mg kg$^{-1}$ and 800 mg kg$^{-1}$ Zn ($P > 0.05$).

**Table 1  Effect of exogenous Zn on the photosynthetic pigments of *P. edulis* Sims f. *edulis* seedlings.**  Different lowercase letters in the same column indicate significant differences between treatments ($P < 0.05$).

| Zinc concentration (mg kg$^{-1}$) | Chl a (mg g$^{-1}$FW) | Chl b (mg g$^{-1}$FW) | Chl a+b (mg g$^{-1}$FW) | Car (mg g$^{-1}$FW) | Chl a/b |
|---|---|---|---|---|---|
| CK | 1.48 ± 0.13b | 0.53 ± 0.04b | 2.01 ± 0.17b | 0.35 ± 0.03b | 2.80 ± 0.02a |
| 200 | 2.04 ± 0.09a | 0.75 ± 0.03a | 2.79 ± 0.12a | 0.45 ± 0.02a | 2.73 ± 0.01b |
| 400 | 1.73 ± 0.09ab | 0.62 ± 0.03b | 2.35 ± 0.13ab | 0.35 ± 0.02b | 2.78 ± 0.00ab |
| 800 | 1.73 ± 0.06ab | 0.62 ± 0.03b | 2.35 ± 0.09ab | 0.40 ± 0.01ab | 2.81 ± 0.02a |

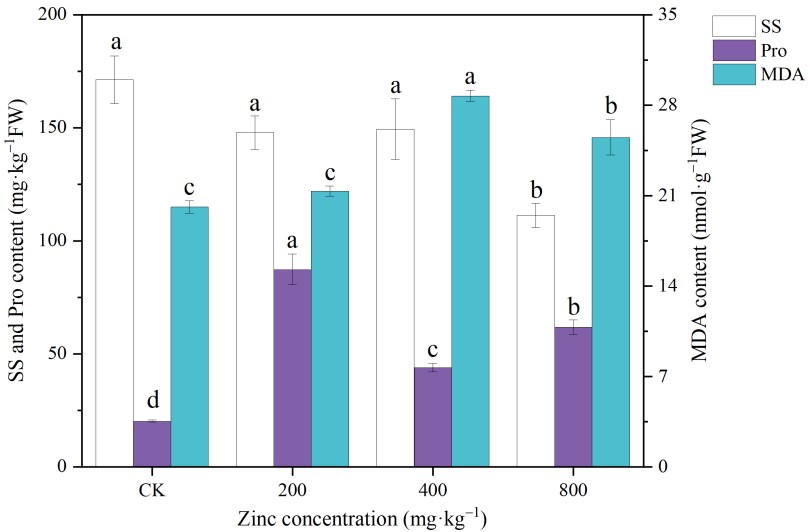

**Figure 2  Effect of exogenous Zn on the SS, Pro and MDA contents of *P. edulis* Sims f. *edulis* seedlings.**

## SS, Pro and MDA contents

SS and Pro are important osmoregulatory substances in plants, and stress leads to changes in the contents of SS and Pro (*Zhang et al., 2018*). As shown in Fig. 2, the SS content of *P. edulis* Sims f. *edulis* seedlings decreased with increasing Zn concentration. The 200 mg kg$^{-1}$ and 400 mg kg$^{-1}$ Zn treatments had no significant effect on the SS content ($P > 0.05$), while the 800 mg kg$^{-1}$ Zn treatment significantly reduced the SS content of *P. edulis* Sims f. *edulis* seedlings (36.67%; $P < 0.05$). The addition of different concentrations of Zn significantly increased the Pro content of *P. edulis* Sims f. *edulis* seedlings by 332.14%, 116.84%, and 202.56%, respectively ($P < 0.05$).

Under stress, unsaturated fatty acids in the protoplasmic membrane of plant cells will be oxidized and produce MDA, which reflects the degree of membrane lipid peroxidation (*Zhang et al., 2018*). As shown in Fig. 2, the addition of 200 mg kg$^{-1}$ Zn had no significant effect on the MDA content of *P. edulis* Sims f. *edulis* seedlings ($P > 0.05$). The addition of 400 mg kg$^{-1}$ and 800 mg kg$^{-1}$ Zn significantly increased the MDA content of *P. edulis* Sims f. *edulis* seedlings (42.69%, 26.7%; $P < 0.05$).

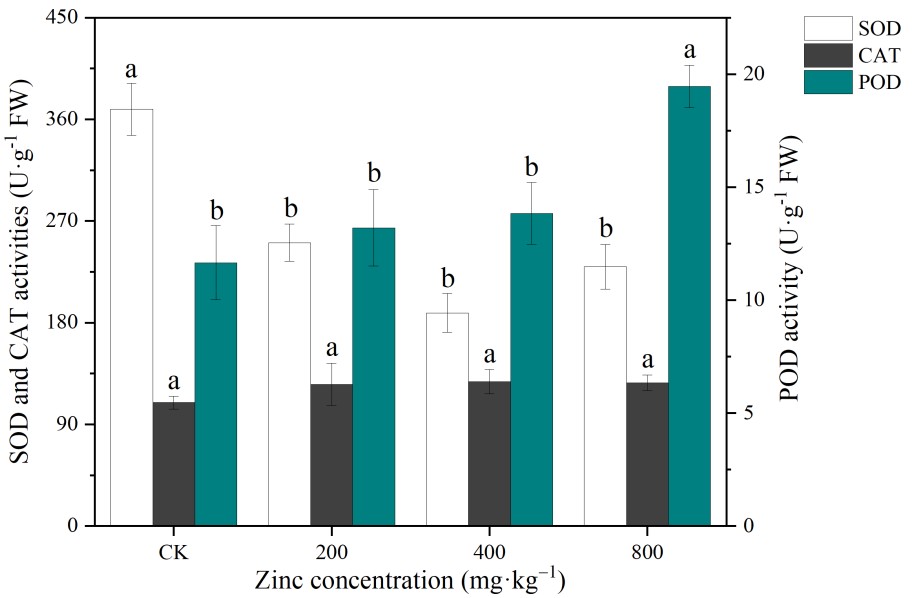

**Figure 3** Effect of exogenous Zn on the CAT, POD and SOD activities of *P. edulis* Sims f. *edulis* seedlings.

## Antioxidant enzyme activity

CAT, POD and SOD are important antioxidant enzymes in plants that can reduce the damage caused by excess free radicals to the cell membranes (*Chowdhury & Choudhuri, 1985*). As shown in Fig. 3, the CAT activity of *P. edulis* Sims f. *edulis* seedlings increased with the addition of exogenous Zn, which increased the POD activity of *P. edulis* Sims f. *edulis* seedlings. With increasing Zn concentration, the POD activity increased gradually, and the POD activity of *P. edulis* Sims f. *edulis* seedlings treated with 800 mg kg$^{-1}$ Zn significantly increased by 67.00% ($P < 0.05$). Compared with CK, the SOD activity of *P. edulis* Sims f. *edulis* seedlings treated with different concentrations of Zn decreased significantly by 30.03%, 47.20% and 32.86% ($P < 0.05$).

## Correlation and principal component analyses

The correlation analysis of the physiological indicators of *P. edulis* Sims f. *edulis* seedlings is shown in Fig. 4A. LA was significantly positively correlated with LP, LL, LW, and SOD. There was a significant negative correlation between LA and MDA, a significant positive correlation between LP and LW and SOD, an extremely significant positive correlation between Chl a, Chl b, Chl a+b, Car and Pro, a significant negative correlation between POD and SS, and a significant negative correlation between SOD and MDA.

Principal component analysis is shown in Fig. 4B. PC1 and PC2 explained 39.60% and 28.80%, respectively, of the variation and accounted for 68.40% of the variation, indicating that the first two principal components reflected the degree of variation in each physiological index. Chl a, Chl b, Chl a+b, Car and Pro together had a large positive load on PC1, indicating that these indexes can be used as important indicators to reflect the growth

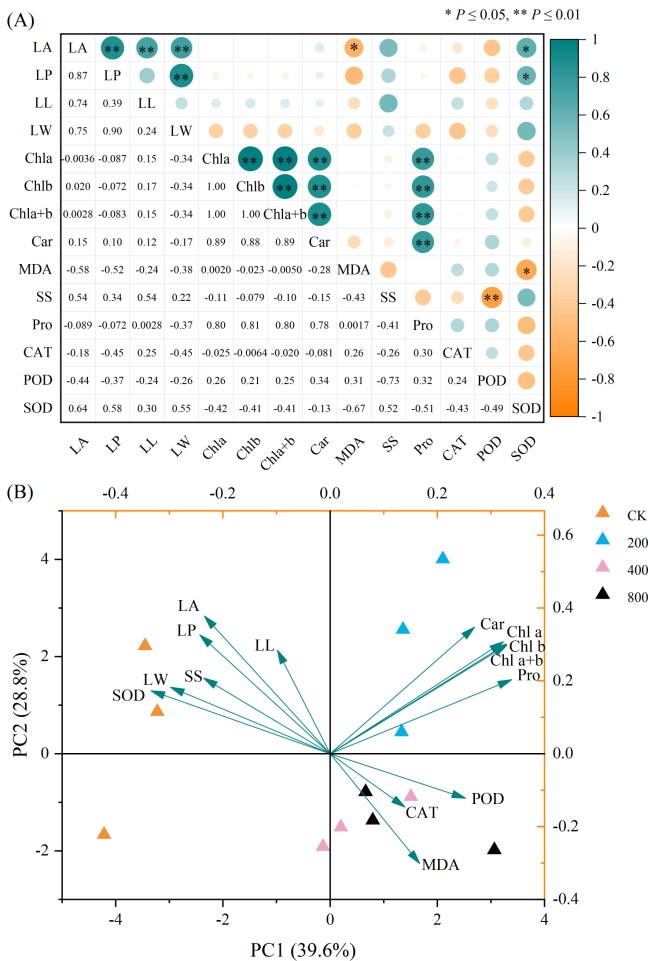

**Figure 4** (A) Correlation and (B) principal component analyses of the physiological indexes of *P. edulis* Sims f. *edulis* seedlings.

status of *P. edulis* Sims f. *edulis* seedlings. SS, SOD, LA, LP, LL and LW had positive loads on PC2, while MDA, CAT and POD had negative loads on PC2. In the principal component analysis, different physiological indexes aggregated to different treatment points, indicating that different concentrations of Zn had different physiological effects on *P. edulis* Sims f. *edulis* seedlings.

# DISCUSSION

Photosynthetic pigments are important physiological parameters in plants; Chl participates in the absorption, transmission and transformation of light energy in the process of photosynthesis, and its content directly reflects the extent of plant photosynthesis (*Chaves, Flexas & Pinheiro, 2009*). Car has a variety of functions in plants and can transfer absorbed light energy to chlorophyll (*Khan et al., 2019*). At the same time, it has the ability to perform antioxidation, scavenge free radicals and protect chlorophyll (*Tracewell et al., 2001*). In this study, the addition of Zn effectively promoted the synthesis of Chl and Car. Because Zn

is one of the constituent structures of Chl, it plays an important role in its structure and function and increases the activity of synthase related to Chl and Car (*Li, Zeng & Su, 2022*), thus promoting the synthesis of both. In this study, the photosynthetic pigment of *P. edulis* Sims f. *edulis* seedlings supplemented with Zn increased first and then decreased, which was similar to the conclusion of *Zhu & Zhou (2016)* and *Li et al. (2007)*. Pro is significantly correlated with photosynthetic pigments, indicating that Pro can affect the synthesis of photosynthetic pigments and then has a certain effect on photosynthesis. The Chl a/b values reflect the stacking degree of thylakoids in chloroplasts. The higher the stacking degree, the more effective the light response (*Zhu & Zhou, 2016*). The Chl a/b values of the *P. edulis* Sims f. *edulis* seedlings decreased first and then increased with increasing Zn concentration, which indicated that the *P. edulis* Sims f. *edulis* seedlings reduced the effect of Zn stress on the light response by regulating the stacking degree of the thylakoids.

SS and Pro play an important role in maintaining the osmotic balance between protoplasts and the external environment, and the accumulation of both can enhance the tolerance of plants to stress (*De Carvalho et al., 2013*; *Zhang et al., 2021*). Meanwhile, SS, as a respiratory matrix, provides energy for various biosynthesis and life activities of crops, while Pro can directly scavenge reactive oxygen species in crop cells and cooperate with the antioxidant system to alleviate oxidative stress (*Song et al., 2013*; *Zhang et al., 2018*). The content of SS decreased significantly and the content of Pro increased significantly in the high-concentration Zn treatment, and there was a significant negative correlation between SS and POD. This result showed that a high concentration of Zn damages the physiological function of *P. edulis* Sims f. *edulis* seedlings and inhibits the synthesis of SS. The *P. edulis* Sims f. *edulis* seedlings consumed SS to provide energy for their antioxidant physiological activities. *P. edulis* Sims f. *edulis* seedlings reduce the water potential and osmotic pressure by synthesizing and accumulating Pro to maintain a normal water supply, which plays an important role in maintaining the original physiological process and protecting cells (*Wang et al., 2016b*). At the same time, Pro can combine with intracellular excess free $Zn^{2+}$ to form a nontoxic complex to alleviate the stress of high concentrations of Zn (*Song et al., 2013*). The increase in Pro may play a key role in resisting the stress of high concentrations of Zn. The increase in the Pro content with the addition of 200 mg kg$^{-1}$ Zn may be caused by other factors in the process of cultivation and management, which need to be further studied.

MDA can react with proteins, nucleic acids, amino acids and other active substances to form insoluble compounds, interfere with the normal metabolic activities of cells and increase the permeability of the cell membrane (*Pauls & Thompson, 1984*). CAT, POD, SOD and other protective substances can maintain the dynamic balance of the production and scavenging of free radicals in plants, thus avoiding oxidative damage to plant cell membranes caused by free radicals (*Li et al., 2007*). Maintaining and increasing the activities of CAT, POD and SOD can be used as the material basis for plant tolerance to heavy metal stress (*Wang et al., 2016a*). In this study, low and medium concentrations of Zn did not cause toxicity to *P. edulis* Sims f. *edulis* seedlings, while medium and high concentrations of Zn caused oxidative damage, increased the cell membrane permeability and affected the normal life activities of the *P. edulis* Sims f. *edulis* seedlings. The activities

of CAT and POD of the *P. edulis* Sims f. *edulis* seedlings were increased by adding Zn, and the antioxidant capacity was enhanced. The activity of POD was still high when treated with high concentrations of Zn, which indicated that the *P. edulis* Sims f. *edulis* seedlings were still tolerant to high concentrations of Zn. The SOD activity of *P. edulis* Sims f. *edulis* seedlings treated with exogenous Zn decreased significantly, which was similar to the results of *Burman, Saini & Praveen (2013)*. It may be that the addition of Zn suppresses the increase in ROS levels, thus reducing the activity of SOD (*Burman, Saini & Praveen, 2013*). The addition of medium and high concentrations of Zn inhibited the activity of SOD because these concentrations also caused oxidative damage to the cell membrane. However, there was a significant negative correlation between SOD and MDA, indicating that SOD was still efficient in scavenging reactive oxygen species, and SOD was closely related to the growth of the *P. edulis* Sims f. *edulis* seedling leaves and played a positive regulatory role.

## CONCLUSION

(1) Exogenous Zn effectively promoted leaf growth and photosynthetic pigment synthesis in *P. edulis* Sims f. *edulis* seedlings, and the 200 mg kg$^{-1}$ Zn treatment had the best effect. (2) The application of 400 mg kg$^{-1}$ and 800 mg kg$^{-1}$ Zn inhibited the synthesis of SS in *P. edulis* Sims f. *edulis* seedlings, reduced the water potential and osmotic pressure through the synthesis and accumulating of Pro to maintain water metabolism and consumed SS to provide energy for antioxidant physiological activities. (3) Zn concentrations of 400 mg kg$^{-1}$ and 800 mg kg$^{-1}$ significantly increased the MDA content of *P. edulis* Sims f. *edulis* seedlings and caused oxidative damage to cell membrane lipids, Zn treatment increased the activities of CAT and POD and enhanced the antioxidant capacity. (4) The correlation between photosynthetic pigments in *P. edulis* Sims f. *edulis* seedlings was very high. Pro affected the synthesis of photosynthetic pigments, and SOD effectively alleviated the peroxidation of cell membrane lipids and was closely related to the growth of the leaves. (5) A suitable concentration of Zn is beneficial to the growth of *P. edulis* Sims f. *edulis* seedlings and imparts a certain tolerance to stress caused by a high concentration of Zn. Therefore, for *P. edulis* Sims f. *edulis*, the concentration of Zn fertilizer should not exceed 200 mg kg$^{-1}$.

### Funding

This work is supported by the Guizhou Science and Technology Support Program (NO. QKHZC (2020) 1Y176, NO. QKHZC (2021) General 460) and the National Key R & D Program of China (2021YFD1100303). The funders had no role in study design, data collection and analysis, decision to publish, or preparation of the manuscript.

### Grant Disclosures

The following grant information was disclosed by the authors:

Guizhou Science and Technology Support Program: QKHZC (2020) 1Y176, QKHZC (2021) General 460.
National Key R & D Program of China: 2021YFD1100303.

## Competing Interests

The authors declare there are no competing interests.

## Author Contributions

- Jianli Zhang conceived and designed the experiments, authored or reviewed drafts of the article, and approved the final draft.
- Tao Yang conceived and designed the experiments, performed the experiments, analyzed the data, prepared figures and/or tables, authored or reviewed drafts of the article, and approved the final draft.
- Chen Zhang performed the experiments, prepared figures and/or tables, and approved the final draft.
- Ting Zhang performed the experiments, prepared figures and/or tables, and approved the final draft.
- Lihua Pu analyzed the data, prepared figures and/or tables, and approved the final draft.
- Weiquan Zhao analyzed the data, prepared figures and/or tables, and approved the final draft.

## Data Availability

The raw data are available in the Supplemental File.

## Supplemental Information

Supplemental information for this article can be found online at http://dx.doi.org/10.7717/peerj.16280#supplemental-information.

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
