# Peer review of "Effects of exogenous zinc on the physiological characteristics and enzyme activities of Passiflora edulis Sims f. edulis seedlings"

_PeerJ, doi:10.7717/peerj.16280_

## Round 0.1 · original submission · Major Revisions

I would encourage authors to revise the manuscript thoroughly and resubmit. Here are a few suggestions from my side.

1. Develop a strong study background and comprehensively analyze the existing research gaps.

2. Make your objectives and tested hypothesis clear. Experimental design can be improved by adding more information on how plant material was collected and treated under controlled conditions. Each method used in M&M should be cited properly. Results should be logically discussed in the discussion section. Don't repeat your results in the conclusion, instead synthesize your results.

Reviewer 1 ·

Basic reporting

The present study "Effects of exogenous Zinc on physiological characteristics and enzyme activities of Passiûora edulis Sims f. edulis" attempts to explore the effects of different concentrations of Zn on photosynthetic pigments, soluble sugar, proline, MDA and enzyme activities of Passiûora edulis Sims f. edulis. The work falls into the scope of the journal.
The English language should be improved to ensure that an international audience can clearly understand your text. Some examples where the language could be improved include lines 42-46, 87-91, 100-102, 115-117, 121-122 and 160-163. – the current phrasing makes comprehension difficult. I would suggest writing short sentences and asking a fluent English-speaking reader to read the manuscript before submission.

Experimental design

Introduction: I suggest the author to rephrase the sentences from line 46 – 56, because they are so consistent.

Materials and methods:
1. Inconsistent capitalization on line 86.
2. The statements on line 87, 89-90 is confusing.
3. How many plants per replication?
4. Why these concentrations of were selected? Why not set up a treatment of 100 mg·kg-1 because the authors mention the background Zn value of 99.5 mg·kg-1 in Guizhou province.
5. How many times has it been treated with exogenous Zn solution? How much each time? What kind of treatment, Leaf spray or watering…? How long did it take? What is the effective duration of Zn?
6. There is a lack of references for the determination of antioxidant enzyme activity.

Results:
1. Since the full names of MDA, Pro, SS, CAT, POD and SOD have been shown in the materials and methods section, there is no need to go into details in the results section.
2. Lack of data or pictures on growth indicators of seedlings treated with exogenous Zn.
3. Lines 168, 171 should be past tense.

Discussion:
1. There are lots of redundancy between the discussion and the result section. One way of improving Discussion is to avoid repetition of results in this part (i.e. line 201-204, 220-228).
2. Please explain why Zn treatment increased CAT and POD activities but decreased SOD activity?
3. Part 4 in Discussion is very shallow and need in depth discussion with the recent literature published.

References:
The format of reference list needs to be further adjusted and unified.

Validity of the findings

The data in the paper are well presented and described. Figures are effective. Conclusions resulting from the figures are valid.
Figure 3 contains two figures, please label A and B.
The discussion section is not written in a very thoughtfully way. There are lots of redundancy between the discussion and the result section.
For further comments refer section Experimental design.

Reviewer 2 ·

Basic reporting

In this manuscript, the authors attempted to assess the physiological responses of Passiûora edulis Sims f. edulis to Zn application. They adopted a pot experiment for monitoring the physiological indices when treated with Zn. I have some concerns:
1)This manuscript needs extensive English polishing.
2)Abstract section: it should be reconstructed, according to a logically manner Background-Method-Results-Conclusion.
3)Background section: this section should reach the purpose and significance of your study.

Experimental design

This section should make clear: How many plants were used for each experiment, and have biological replicates included?

Validity of the findings

1)Conclusion section: it is not repeat of your results.
2)The figure legends are not clear, which should let the figure stand alone.

---

## Round 0.2 · Minor Revisions

Please thoroughly revise your manuscript for grammatical and other mistakes.

**Language Note:** The Academic Editor has identified that the English language must be improved. PeerJ can provide language editing services - please contact us at copyediting@peerj.com for pricing (be sure to provide your manuscript number and title). Alternatively, you should make your own arrangements to improve the language quality and provide details in your response letter. – PeerJ Staff

Reviewer 1 ·

Basic reporting

No comment

Experimental design

No comment

Validity of the findings

No comment

Reviewer 2 ·

Basic reporting

Although the authors made some revisions and claimed that they had the version edited by English-speaking experts, the background part still needs to be strengthened to logically make the objective clearly and the language should be improved.

Experimental design

This part is ok.

Validity of the findings

This part is ok.

---

## Round 0.3 · accepted · Accept

The authors have addressed all of the reviewers' comments. This manuscript should be accepted now for publication.